# New Record of Hydrothermal Vent Squat Lobster (*Munidopsis lauensis*) Provides Evidence of a Dispersal Corridor between the Pacific and Indian Oceans

Hee-seung Hwang [1], Boongho Cho [2,3], Jaemin Cho [3], Beomseok Park [4,*] and Taewon Kim [2,3,*]

1 Research Institute of EcoScience, Ewha Womans University, Seoul 03760, Korea; winsome212@nate.com
2 Program in Biomedical Science and Engineering, Inha University, 100 Inha-ro, Incheon 22212, Korea; boonghocho@gmail.com
3 Department of Ocean Sciences, Inha University, 100 Inha-ro, Incheon 22212, Korea; cjm970131@naver.com
4 Department of Biomedical Laboratory Science, College of Health Science, Eulji University, Seongnam 13135, Korea
* Correspondence: bspark74@eulji.ac.kr (B.P.); ktwon@inha.ac.kr (T.K.); Tel.: +82-10-8726-3070 (T.K.)

**Abstract:** Hydrothermal vents are chemosynthetically driven ecosystems and one of the most extreme environments on Earth. Vent communities exhibit remarkable taxonomic novelty at the species and supra-species levels, and over 80% of vent species are endemic. Here, we used mitochondrial DNA to identify the biogeographic distribution of *Munidopsis lauensis* and the heme-binding regions of A1-type COX1 from six species (including *M. lauensis*) to investigate whether genetic variation in the protein structure affects oxygen-binding ability. We verified the identity of Indian Ocean specimens by comparing sequences from the barcoding gene mitochondrial cytochrome oxidase subunit 1 (COI) with known *M. lauensis* sequences from the NCBI database. The data show that these are the first recorded specimens of *M. lauensis* in the Indian Ocean; previously, this species had been reported only in the southwest Pacific. Our findings support the hypothesis that vent fauna in the Pacific and Indian Oceans can interact via active ridges. In the case of the mitochondrial DNA-binding site, the arrangement of heme-binding ligands and type A1 motif of *M. lauensis* was identical to that in other species. Moreover, our findings suggest that the mechanism of oxygen binding is well conserved among species from terrestrial organisms to hydrothermal extremophiles. Overall, dispersal of the same species to geologically separated hydrothermal vents and conserved heme-binding regions in mitochondrial proteins suggest that hydrothermal species might have evolved from shallow sea organisms and became distributed geographically using a dispersion corridor.

**Keywords:** hydrothermal vents; Munidopsis; dispersal corridor; vent fauna; heme-binding site

## 1. Introduction

Chemosynthetically driven hydrothermal vents are among the harshest ecosystems on the planet, having temperatures as high as 390 °C, low oxygen levels, and enriched concentrations of hydrogen sulfide ($H_2S$), methane ($CH_4$), and heavy metals, including iron, zinc, and copper [1]. This unique environment makes vent species susceptible to geological settings and local ecosystems because they produce chemical-rich fluids that support the food web, generating a patchwork seafloor habitat. Catastrophic disturbances to vent species or vents can eradicate entire communities [2]. The characteristics of vent communities are complex, and although 82% of vent species are endemic, they are remarkably diverse at multiple taxonomy levels (e.g., family, order, class, and species) [3].

Hydrothermal vents are found at mid-ocean spreading centers in the Atlantic, Arctic, Indian, and eastern Pacific Oceans, and back-arc basins of the western Pacific Ocean [4]. In particular, numerous vent communities have been reported along active margins in the Atlantic [5,6], Pacific [7,8], and Indian Oceans [9,10].

Phylogeography is a critical ecological characteristic that explains a species' evolutionary history and successful adaptation to environmental changes. In terms of community, most organisms in Indian Ocean vent fields have evolutionary relationships with western Pacific vent fauna [10]. However, exceptions exist, such as the shrimp *Rimicaris aff. exoculata*, a decapod that is the predominant species in Indian Ocean vents but is similar to its mid-Atlantic counterpart [10]. Global assessments of chemosynthetic faunal biogeography have suggested that the Indian Ocean vent communities follow asymmetric assembly rules biased toward Pacific evolutionary alliances [10].

Decapods (e.g., alvinocaridid shrimp, bythograeid crabs, and galatheid squat lobsters) represent approximately 10% of taxa in hydrothermal vents and are the dominant fauna [1,11,12]. Genetics appear to have participated in bythograeid crab and alvinocaridid shrimp adaptation to vent habitats [13,14]. Munidopsis is the second-largest and ecologically diverse genus of galatheid squat lobsters. Globally, more than 200 species have been defined: more than 150 species in the Indo-Pacific and at least 70 in the Atlantic [15,16]. Although only 10 species are endemic to hydrothermal vent environments [17–22], they exhibit a unique pattern of distribution and abundance. Five species are found in the western Pacific (*M. starmer*, *M. sonne, M. lauensis, M. marianica*, and *M. myojinensis*), three in the eastern Pacific (*Munidopsis* sp., *M. subsquamosa*, and *M. lentigoI*), and two in the Mid-Atlantic Ridge (*M. exuta* and *M. acutispina*) [22]. Interestingly, all Munidopsis species have limited distributions, comprising three coexisting species at most (Table 1).

**Table 1.** Diversity of *Munidopsis* in hydrothermal vent.

| Ocean | Hydrothermal Vent | Depth | Species | Reference |
|---|---|---|---|---|
| **Western Pacific Ocean** | Brother Seamount (34°51′45.4″ S, 179°03′28.6″ E) | 1649–1750 m<br>1649–1992 m<br>1649 m | *M. lauensis*<br>*M. sonne*<br>*M. kermadec* | [23] |
| | North Fiji Basin (16°59′00.0″ S, 173°55′00.0″ E) | no record | *M. lauensis*<br>*M. sonne*<br>*M. starmer* | [21] |
| | Lau Basin (20°59′21.0″ S, 176°34′06.0″ W) | 1750 m | *M. lauensis* | [21,24] |
| | Izu-Bonin arc (32°06.25′ N, 139°52.17′E) | 1288–1625 m | *M. myojinensis* | [21] |
| | Manus Basin (3°43′49.4″ S, 151°40′27.5″ E) | no record | *M. lauensis* | [21] |
| | Mariana (18°11′00.0″ N, 144°45′00.0″ E) | no record | *M. marianica* | [21] |
| | Formosa Ridge (22°06′54.0″ N 119°17′06.0″ E) | 1750–2000 m | *M. lauensis* | [25] |
| **Indian Ocean** | **Onnuri Vent Field(11°24′54.9″ S, 66°25′24.0″ E)** | **2023 m** | ***M. lauensis*** | **the present study** |
| | Kairei Field (25°19′14.4″ S, 70°02′24.0″ E) | 2422–2435 m | *M. larticorpus* | [22] |
| | Forecast Vent Field/Mariana Back Arc Basin (13°23.07′ N 145°55.02′ E) | 1450 m | *M. gracilis* | [22] |
| **Mid-Atlantic Ridge** | Mid-Atlantic Ridge (1°38′50.7″ N 19°42′41.1″ W) | no record | *M. acutispina* | [22] |
| | Mid-Atlantic Ridge (2°18′10.9″ N 25°38′33.9″ W) | no record | *M. exuta* | [21,26] |
| | East Pacific Rise S of Baja California (20°49′36.0″ N 109°06′00.0″ W) | 3502 m | *M. lentigo* | |
| **Northern Pacific Ocean** | Galapagos Rift (eastern Pacific 13° N and 21° N areas) | no record | *M. subsquamosa* | [25] |
| | Limbo Vent, Juan De Fuca Ridge (46°00′02.0″ N 129°59′59.1″ W) | 1545–2008 m | *M. alvisca* | [17] |

Not all vent fauna have distributions as restricted as Munidopsis. Molecular comparisons have revealed genetic affinity between the following species in the Indian and western Pacific Oceans: shrimp *Rimicaris kairei*, gastropod Alviniconcha sp., bythograeid

crab *Austinograea rodriguezensis*, stalked barnacle *Neolepas* sp., deep-sea mussel *Bathymodiolus marisindicus*, sea anemone Marianactis sp., and scaly foot gastropod [10,27].

Mitochondria are responsible for most cellular aerobic metabolism, producing ATP through the electron transport chain. All 13 mitochondrial protein-coding genes are involved in this process. Functional restrictions on mitochondrial genes are related to the following adaptive evolutionary mechanisms: climatic adaptation [28], locomotion [28,29], high elevation adaptation (low-oxygen and cold climate) [30–35], mammalian adaptation [36], and deep-sea hydrostatic pressure adaptation [37]. DNA barcoding with a cytochrome c oxidase subunit 1 (COX1) region has been widely used to recognize species in taxonomic studies [38]. The COX1 region is divided into three evolutionary families: type A (mitochondrial-like oxidases), B (ba$_3$-like oxidases), and C (cbb$_3$-type oxidases). The structural diversity of these families correlates with different proton pumping efficiencies [39].

In this study, we investigated the biogeographic distribution of *Munidopsis lauensis* and tested a hypothesis that the Indian Ocean is a dispersal corridor connecting the hydrothermal vent fauna of the Atlantic and Pacific Oceans. We also compared the heme-binding regions of proteins from six species, including *M. lauensis*, to evaluate whether genetic variation affects oxygen-binding ability. Overall, the purpose of this study was to obtain information about the evolutionary history of *M. lauensis*, including its dispersal mechanisms, using mitochondrial data.

## 2. Materials and Methods

### 2.1. Sampling and Mapping

On 1 July 2019, eighteen samples of *M. lauensis* were collected at the Onnuri Vent Field (OVF; latitude: 11°14′58.6″ S; longitude: 66°15′14.4″ E) in the Indian Ocean using a TeleVision-grab (TV-grab) device mounted on the Research Vessel (R/V) ISABU (Dive No. 6). Samples were immediately fixed in 75% ethanol (Figure 1). The sampling depth was approximately 2023.2 m.

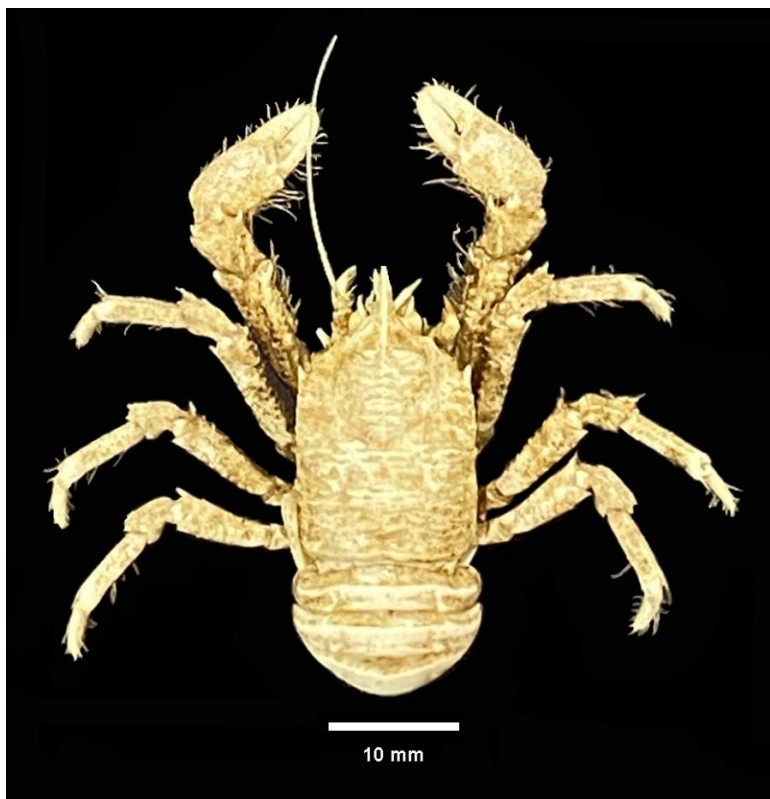

**Figure 1.** *Munidopsis lauensis* Baba and de Saint Laurent, 1992.

A map (Figure 2A) was developed using a Diva-Gis 7.5 template (http://www.diva-gis.org/ accessed on 3 January 2022). Sampling locations of the specimens included in the analyses and data from Major Ocean Currents (2016) and Global Distribution of Hydrothermal Vent Fields (2020) in ArcGIS were combined.

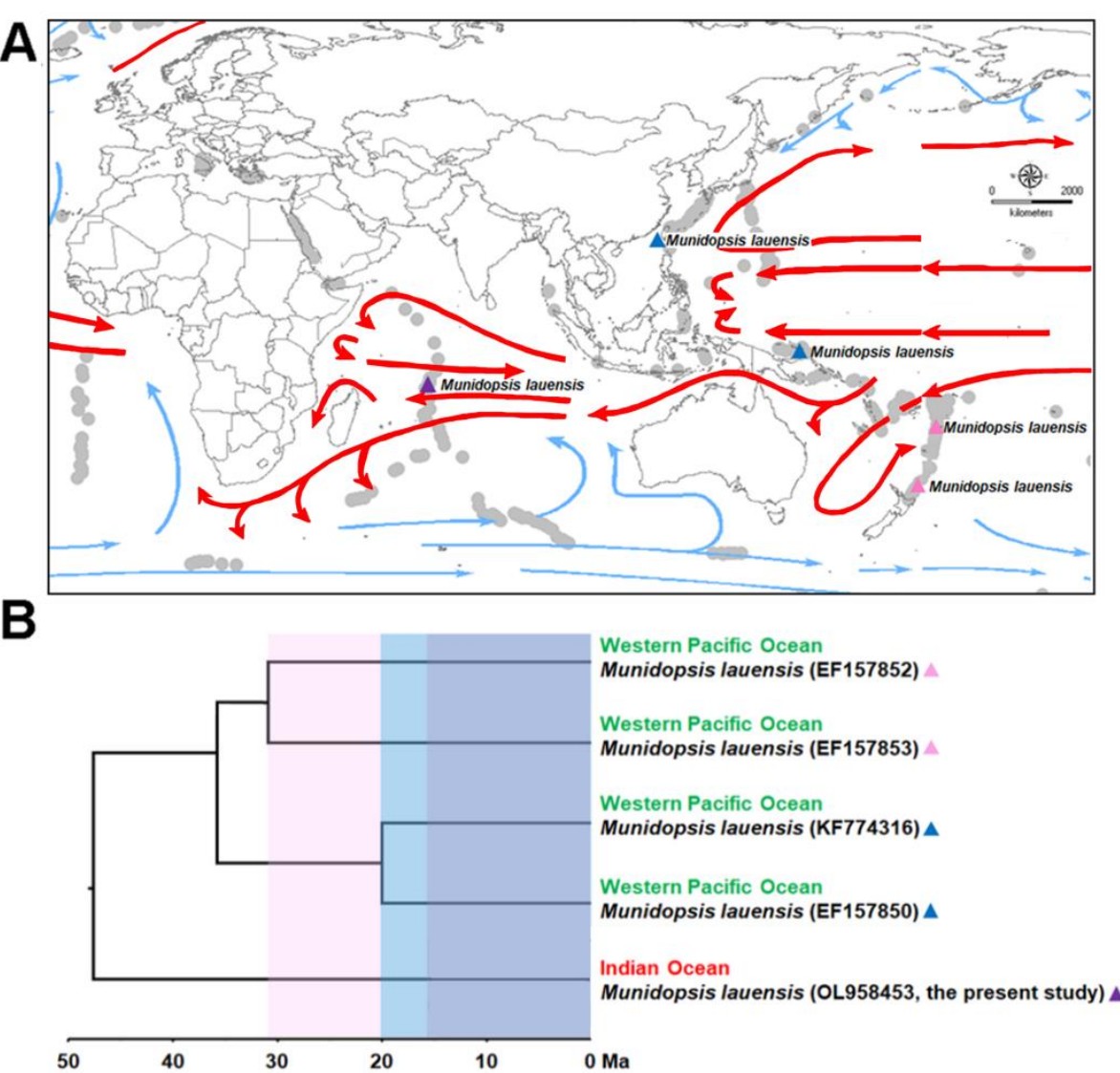

**Figure 2.** (**A**) Biogeographic distribution of *M. lauensis*. Red arrows indicate warm currents and blue arrows indicate cold currents. (**B**) Schematic diagram of divergence analysis for *M. lauensis*.

### 2.2. DNA Barcoding

Genomic DNA was extracted from the muscle tissue of five *M. lauensis* specimens using a Qiagen DNeasy Blood and Tissue kit. The barcoding regions of the mitochondrial cytochrome oxidase subunit 1 (COI) genes (658 bp) were sequenced and compared with published sequences from western Pacific Ocean isolates (accession numbers: EF157850–EF157853) and *M. verrilli* Benedict, 1902. New sequences from Indian Ocean specimens were registered in GenBank (accession numbers: OL958453–OL958457). *Shinkaia crosnieri* Baba and Williams, 1998 (accession numbers: MK795354–MK795356) were selected as closely related members of Munidopsidae based on the complete mitochondrial genome [40]. *Munida gregaria* (Fabricius 1793) (accession numbers: NC030255, KU521508) was selected because Mundidae and Mundopsidae are phylogenetically related families [40]. *Munida gregaria* and *Shinkaia crosnieri* were used as outgroups.

Primers used to amplify COI-5 by PCR were LCO1490 (5′-GGT CAA ATC ATA AAG ATA TTG G-3′) and HCO2198 (5′-TAA ACT TCA GGG TGA CCA AAA AAT CA-3′) [41]. The following thermocycling program was used: 5 min at 94 °C; 40 cycles of 1 min at 94 °C, 1 min at 40 °C, and 2 min at 72 °C; and a final extension at 72 °C for 10 min. The 25 µL reaction mix included 15.7 µL ultrapure water, 5 µL of 5X PCR buffer, 2 µL of each primer (10 µM), 1 µL of dNTP (10 mM), 0.3 µL of Taq polymerase (5 U), and 1 µL of DNA template. Sequences were aligned using MAFFT [42]. Sequence divergence between individuals was quantified using the Kimura 2-parameter (K2P) distance model [43]. A neighbor-joining (NJ) tree of K2P distances was created in MEGA X [44].

### 2.3. Estimation of Divergence Time

The most appropriate model to estimate divergence time was the HKY+Gamma model, which was selected using PartitionFinder version 1.1.1, Australia [45]. The Bayesian phylogenetic software BEAST version 2, USA [46] was used to estimate the divergence time. Analyses were performed using a strict clock and an uncorrelated lognormal relaxed molecular clock to check for rate variation among branches [47]. We conducted analyses using a Yule speciation model and a birth-death model for the tree prior to evaluate whether the sensitivity of the results was affected by the choice of tree prior. Posterior distributions of parameters were estimated using Markov chain Monte Carlo (MCMC) sampling over $10^8$ steps, with samples drawn every $10^4$ steps. The initial 10% of samples were discarded as burn-in. Convergence was checked by running the analysis in duplicate and visualizing the results in the program Tracer version version 1.7.1, USA [48], which showed that the effective sample size of all parameters was above 200. We used Tree Annotator version 1.8.4, USA [49], available in the BEAST version 2, USA package, to identify the maximum clade credibility tree.

### 2.4. Heme-Binding Site of COX1 Sequences

We used COX 1 sequences from five species classified as "type of COX1" to analyze the heme-binding region in *M. lauensis* sequences. Sequences from *Homo sapiens*, *Bos taurus*, *Escherichia coli* bo3, *Drosophila melanogaster*, and *Portunus trituberculatus* were collected from NCBI (https://www.ncbi.nlm.nih.gov/ accessed on 3 January 2022) and compared with the *M. lauensis* data. The heme-binding site sequences were aligned by multiple sequence alignment using Clustal Omega (https://www.ebi.ac.uk/Tools/msa/clustalo/ accessed on 3 January 2022) [50]. In this comparison, we focused on whether each species contains the same region found in human COX1. Secondary structures based on that of human COX1 were analyzed by comparing amino acids using ESPript 3.0 (https://espript.ibcp.fr/ESPript/ESPript/ accessed on 3 January 2022) [51]. The model structure of *M. lauensis* COX1 was calculated using a SWISS-MODEL (https://swissmodel.expasy.org/ accessed on 3 January 2022) [52]. The structural data of Human COX1 (PDB code: 5z62) and *M. lauensis* COX1 were described in PyMOL (https://pymol.org/2/ accessed on 3 January 2022) [53]. The superposition showed that the two proteins have secondary structure alignment and helped to identify heme-binding sites and type A1 motifs in *M. lauensis* COX1.

## 3. Results and Discussion

### 3.1. Biogeographic Distribution of M. lauensis

Indian Ocean *M. lauensis* specimens (accession numbers: OL958453–OL958457) have COI-barcoding regions nearly identical to West Pacific specimens (Brothers Seamount, Manus, and Lau Basins) (accession numbers: EF157850, 1587852, and 157853) [21], with a maximum of 0.58% sequence divergence (Figure 3). We conclude that our collected specimens are indeed *M. lauensis* and represent the first records of this species in the Indian Ocean. All other isolates of *M. lauensis* were from deep-sea hydrothermal vents in the southwest Pacific Ocean [54].

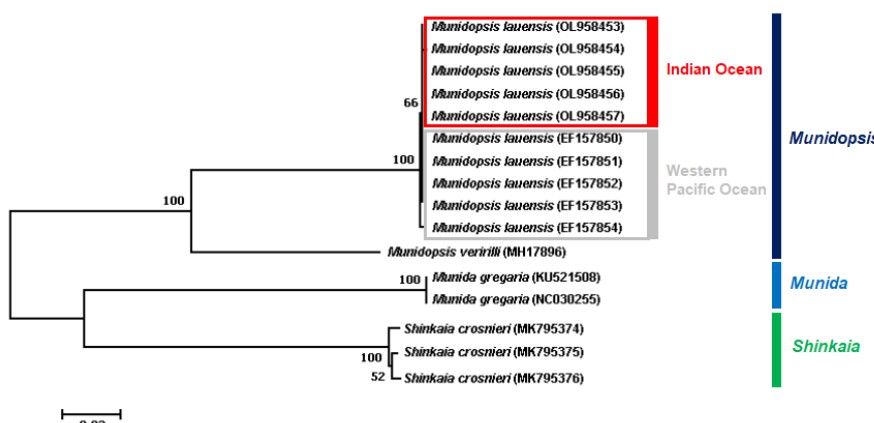

**Figure 3.** Genetic relationships between *Munidopsis lauensis* isolates from the Indian and western Pacific Oceans based on mitochondrial cytochrome oxidase subunit 1 (COI) sequences. *Munida gregaria* and *Shinkaia crosnieri* were used as outgroups.

The presence of this species in different oceans supports the hypothesis that the Indian Ocean is a dispersal corridor for hydrothermal vent fauna between the Atlantic and Pacific Oceans. Indeed, other faunal and molecular comparisons have also revealed an affinity between taxa in the Indian and western Pacific Oceans [10] (Figure 2A).

The order of divergence times was as follows: First, among the western Pacific *M. lauensis* isolates, the sequences from the Hine Hina material (accession number: EF157852) and the Brothers Seamount material (GenBank number: EF157853) were nearly identical and first diverged at 30 Myr. Second, the sequences from the Desmos material (accession number: EF157850) and Taiwan (accession number: KF774316) diverged at 20 Myr. Lastly, our material from the Indian Ocean (accession numbers: OL958453–OL958454) diverged at approximately 15 Myr (Figure 2B). Although it is impossible to review all distributions of the present species *M. lauensis*, this study reveals that the western Pacific group diverged before the Indian Ocean group.

Our study is the first to suggest that a dispersal corridor exists between the western Pacific and Indian Oceans, based on the occurrence of *M. lauensis* in both regions. Furthermore, our data are aligned with a previous study demonstrating the existence of a dispersal corridor between the Pacific and Atlantic Oceans [23] and therefore supports the hypothesis that dispersal corridors for hydrothermal vent species can reside in different oceans. Larvae of hydrothermal vent species floating on the ocean surface might have spread on ocean currents and then have settled onto various hydrothermal vents [55]. Although the precise dispersal mechanism is unknown, even if the probability that floating larvae could successfully settle in different locations is low, it could still have happened.

### 3.2. Binding Sites of COX in M. lauensis

COI proteins are categorized into three evolutionary families: Type A (mitochondrial-like oxidases), B (ba3-like oxidases), and C (cbb3-type oxidases). The structural diversity of COX1 reflects differences in proton pumping efficiency [39]. The protein sequence of type A COX1 comprises the motif -GHPEVY-. The helix 6 residues divide this motif into two subfamilies: type A1 (glutamate residue in the motif -XGHPEV-) and type A2 (tyrosine and serine in the alternative motif -YSHPXV-). The type A1 motifs of *M. lauensis* and other species are homologous (Figure 4). The heme-binding sites of human COX1 are known as H61, H240, H290, H291, H376, H378, and Y244 [56]. Seven heme-binding amino acids are arranged on the same helix in all species. This analysis shows that even if organisms have evolved in different environments, the arrangements of heme-binding ligands and type A1 motifs remain unchanged in *M. lauensis* and other species. Type A1 COX1 regions are highly conserved, having practically identical structures across numerous species (Figure 5).

However, proton transport efficiency can vary among organisms. More research is needed to evaluate differences in respiratory efficiency between *M. lauensis* COX and other species.

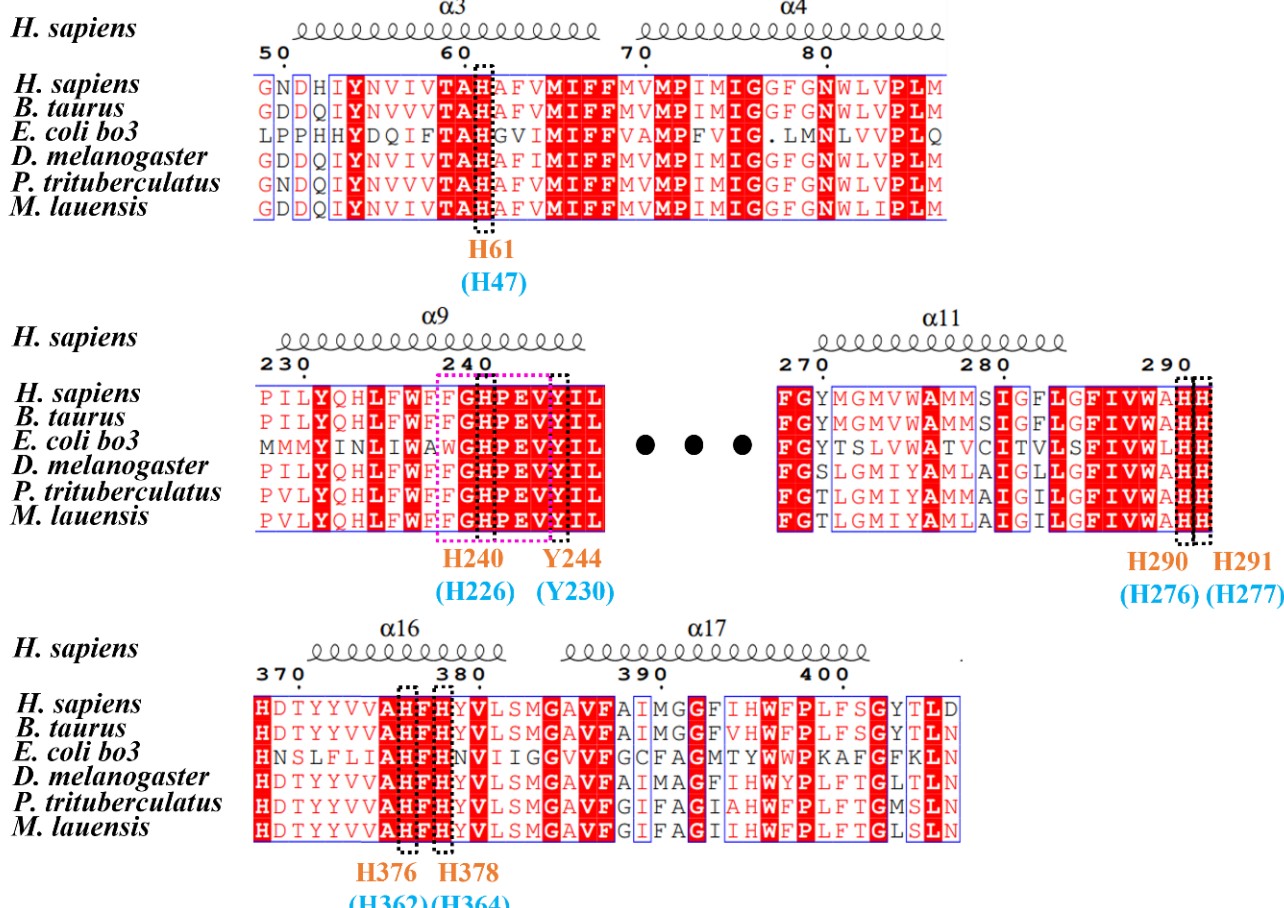

**Figure 4.** Sequence alignment of the COX1 type A1 motif in six species (accession numbers in parentheses). *H.*, *Homo* (AGW78696), *B.*, *Bos* (P00396), *E.*, *Escherichia* (P0ABI8), *D.*, *Drosophila* (AAF77227), *P.*, *Portunus* (QPD06751), *M.*, *Munidopsis* (QFG40073). Red boxes and white characters indicate strict sequence identities, and red characters show residues with high similarity. Black-dashed boxes and orange labels indicate the heme-interacting residues of human COX1 (H61, H240, Y244, H290, H291, H376, and H378). Cyan labels represent the H47, H226, Y230, H276, H277, H362, and H364 residues of *M. lauensis* COX1. Magenta-dashed boxes represent the COX1 type A1 motif (-XGHPEV-). Secondary structures are represented above the alignment.

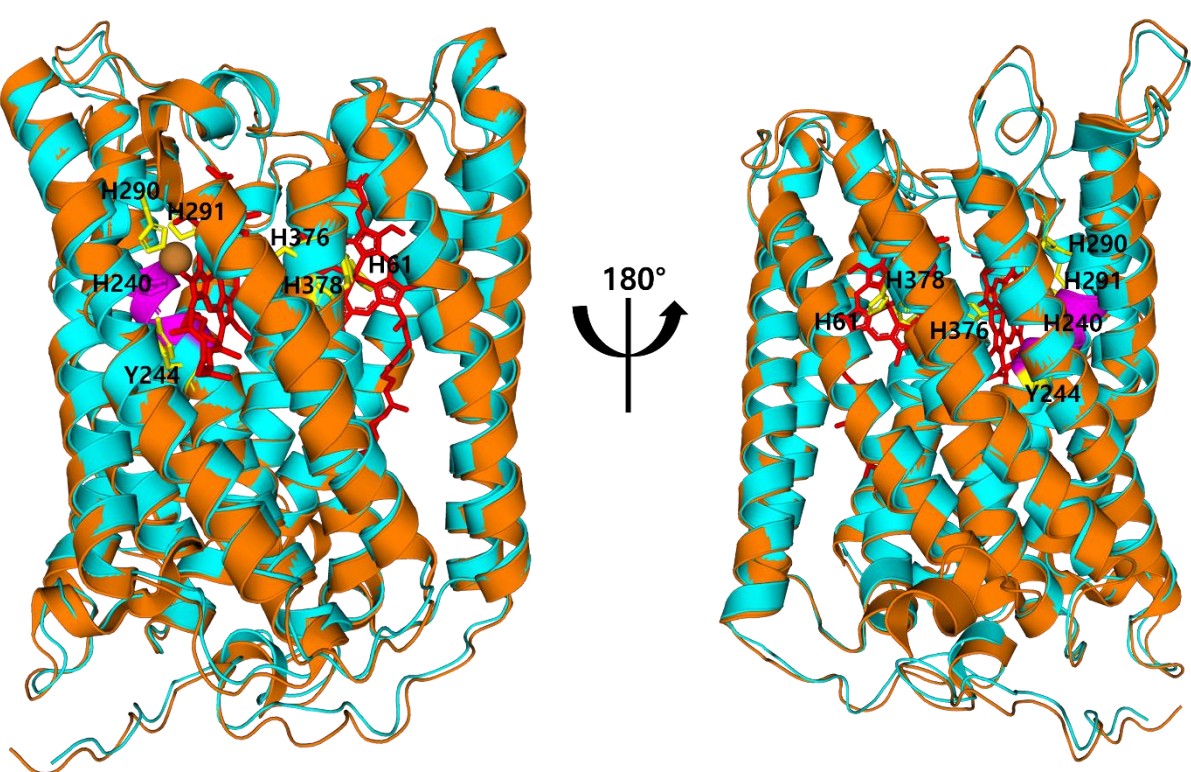

**Figure 5.** Comparison of predicted structure models for *M. lauensis* COX1 and human COX1 (PDB: 5z62). Human and *M. lauensis* COX1 are colored in orange and cyan, respectively. Heme-binding amino acids (yellow), heme (red), Cu$_B$ (brown), and the type A1 motif (magenta) are also shown. The type A1 motif of helix 9 comprises -XGHPEV- amino acid sequences in all species. The heme-binding regions of *M. lauensis* and five species have the same location as the human protein. Indeed, amino acid sequences around the heme-binding regions of all species are highly homogeneous. The type A1 motif is strictly conserved between human and *M. lauensis*.

## 4. Conclusions

Mitochondria are predominantly responsible for aerobic metabolism, and functional restrictions relate to adaptive evolutionary mechanisms. Our findings indicate that our isolates of the decapod crustacean *M. lauensis* from a deep-sea hydrothermal vent in the Indian Ocean are new. Previous records were from vents of the southwest Pacific. In addition, the arrangement of heme-binding regions and type A1 motifs of *M. lauensis* are identical to those of six other species. Dispersal of the same species to geologically separated hydrothermal vents with conserved heme-binding regions in mitochondria suggest that hydrothermal species might have evolved from shallow sea environments and a dispersal corridor. Our study provides useful information for additional studies on hydrothermal vents. Furthermore, our study guides future work to characterize the dispersal corridor between the western Pacific and Indian Oceans.

**Author Contributions:** Conceptualization, H.-s.H., B.P. and T.K.; methodology, H.-s.H., B.C. and J.C.; investigation, B.C. and J.C.; data curation, H.-s.H., B.C. and J.C.; writing—original draft preparation, H.-s.H., B.C., J.C., B.P. and T.K.; writing—review and editing, B.P. and T.K.; visualization, H.-s.H., J.C., B.P. and T.K.; project administration, B.C. and T.K.; funding acquisition, T.K. All authors have read and agreed to the published version of the manuscript.

**Funding:** This research was a part of the project titled "Understanding the deep-sea biosphere on seafloor hydrothermal vents in the Indian Ridge" (No. 20170411) funded by the Ministry of Oceans and Fisheries, Korea. It was also partly funded by the Korea Institute of Ocean & Technology (KIOST) research program (Grant no. PE99985). B. Park was supported by Eulji University in 2020.

**Institutional Review Board Statement:** Not applicable.

**Informed Consent Statement:** Not applicable.

**Data Availability Statement:** All data used are presented in the text, figures, and tables.

**Acknowledgments:** We are grateful to Ok Hwan Yu and Dong Sung Kim for helping collection of samples.

**Conflicts of Interest:** The authors declare no conflict of interest.

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
