# Peer review of "New Record of Hydrothermal Vent Squat Lobster (Munidopsis lauensis) Provides Evidence of a Dispersal Corridor between the Pacific and Indian Oceans"

_jmse, doi:10.3390/jmse10030400_

Round 1
Reviewer 1 Report
The manuscript reports on the discovery of the squat lobster Munidopsis laurensis from hydrothermal vent complexes in the Indian Ocean. The species had previously only been described from vent communities in the western Pacific Ocean, which suggests a potential corridor for vent species between these two ocean basins that combines with previous evidence for linkage to the Atlantic Ocean. Genetic analysis confirmed that the Indian Ocean specimens were M. laurensis, which previous studies suggest is one of the most widely distributed squat lobster species in the western Pacific. In addition to providing evidence for linkage for hydrothermal vent communities between the two ocean basins, the authors compared heme-binding regions of AI-type COX1 from six widely divergent species. The results suggest that the region is highly conserved across all groups and likely supports the hypothesis that vent organisms are derived from shallow water marine species.
The manuscript is very well written and contains new and relevant information. The literature review is thorough, the goals are clearly stated and the methods are easily understood. The results are significant in terms of extending the range of a vent species to a new ocean basin suggesting linkage between basins. The other relevant finding is the conservation of genetic structure of mitochondrial genes responsible for aerobic respiration across the range of living organisms from bacteria to mammals.
The authors have submitted a very welldone manuscript that contains both interesting and important new information across several fields. I support acceptance of the manuscript and have no suggestions for revisions.

Author Response
Thank you very much for your review of understanding and judging our manuscript accurately.
Reviewer 2 Report
Two small corrections: first reference line spacing is off in my copy
Reference 17: italics on Galacantha
A well-presented paper. I look forward to seeing it published.
Author Response
Thank you for your detailed review. We modified the line spacing of the first reference to be the same as the line spacing of the other references. Also changed the corresponding genus name to italics.
Reviewer 3 Report
Congratulations for this nice and interesting contribution. No further changes needed, you did a good job.
Author Response
Thank you for reviewing our manuscript and for accepting it without further revision.